# Diagnosis and Management of Pulmonary NTM with a Focus on *Mycobacterium avium* Complex and *Mycobacterium abscessus*: Challenges and Prospects

**DOI:** 10.3390/microorganisms11010047

**Published:** 2022-12-23

**Authors:** Christian Hendrix, Myah McCrary, Rong Hou, Getahun Abate

**Affiliations:** Department of Internal Medicine, Saint Louis University, St. Louis, MO 63104, USA

**Keywords:** NTM, MAC, MAB, *Mycobacterium*

## Abstract

Background: Nontuberculous mycobacteria (NTM) are ubiquitous. NTM can affect different organs and may cause disseminated diseases, but the pulmonary form is the most common form. Pulmonary NTM is commonly seen in patients with underlying diseases. Pulmonary *Mycobacterium avium* complex (MAC) is the most common NTM disease and *M. abscessus* (MAB) is the most challenging to treat. This review is prepared with the following objectives: (a) to evaluate new methods available for the diagnosis of pulmonary MAC or MAB, (b) to assess advances in developing new therapeutics and their impact on treatment of pulmonary MAC or MAB, and (c) to evaluate the prospects of preventive strategies including vaccines against pulmonary MAC or MAB. Methods: A literature search was conducted using PubMed/MEDLINE and multiple search terms. The search was restricted to the English language and human studies. The database query resulted in a total of 197 publications. After the title and abstract review, 64 articles were included in this analysis. Results: The guidelines by the American Thoracic Society (ATS), European Respiratory Society (ERS), European Society of Clinical Microbiology and Infectious Diseases (ESCMID), and Infectious Diseases Society of America (IDSA) are widely applicable. The guidelines are based on expert opinion and there may be a need to broaden criteria to include those with underlying lung diseases who may not fulfill some of the criteria as ‘probable cases’ for better follow up and management. Some cases with only one culture-positive sputum sample or suggestive histology without a positive culture may benefit from new methods of confirming NTM infection. Amikacin liposomal inhalation suspension (ALIS), gallium containing compounds and immunotherapies will have potential in the management of pulmonary MAC and MAB. Conclusions: the prevalence of pulmonary NTM is increasing. The efforts to optimize diagnosis and treatment of pulmonary NTM are encouraging. There is still a need to develop new diagnostics and therapeutics.

## 1. Background

Nontuberculous mycobacteria (NTM) are vast and include all mycobacteria except *M. tuberculosis* (Mtb) and *M. leprae*. Nontuberculous mycobacteria (NTM) have a lipid-rich outer membrane generating hydrophobicity and can produce biofilms, thus allowing a wide variety of hosts within nature [1].

NTM are ubiquitous in nature. Currently, over 190 species of nontuberculous mycobacteria (NTM) have been identified [2]. It is believed that NTM infections are acquired from the environment via inhalation, ingestion and skin contact, which may result in pulmonary disease, lymphadenitis, skin and soft tissue infections, or disseminated disease. Pulmonary disease is the most common form of NTM disease in patients who are negative for human immunodeficiency virus (HIV) [3]. Pulmonary NTM commonly occurs in patients with underlying lung diseases [4,5]. Disseminated diseases are commonly seen in patients who are immunocompromised including HIV, immunosuppressive medications, and genetic defects in Th1 immune responses [6].

In developed countries where tuberculosis (TB) is well controlled, the prevalence of pulmonary NTM and the mortality rate associated with NTM infections are increasing [3,7,8,9,10,11]. A US study of Medicare Part B beneficiaries showed that the prevalence of NTM increased from 20 to 47 per 100,000 persons between 1997 and 2007, an 8.2% increase per year [7]. A more recent report estimated that the number of pulmonary NTM cases in the US increased by at least two-fold between 2010 and 2014 [12]. Data on NTM isolates from pulmonary samples obtained from 30 countries across six continents in 2008 showed that *M. avium* complex (MAC) constituted 37% of isolates in Europe, 52% of in North America and 71% in Australia [13]. In North America, recent data suggest that MAC and *M. abscessus* (MAB) are common causes of pulmonary NTM [14]. Pulmonary MAB is associated with treatment failure rates exceeding 50%, a rapid decline in lung function, significant morbidity and mortality [15,16]. Therefore, this review focuses on pulmonary MAC and MAB, and is prepared with the following objectives: (a) evaluate new methods available for the diagnosis of pulmonary MAC or MAB, (b) assess advances in developing new therapeutics and their impact on treatment of pulmonary MAC or MAB, and (c) evaluate the prospects of preventive strategies including vaccines against pulmonary MAC or MAB.

## 2. Methods

A literature search was conducted using PubMed/MEDLINE and multiple search terms. The search included publication dates 1 January 1980–16 December 2022 and was restricted to the English language and human studies. A search term ‘pulmonary NTM and new methods’ revealed 116 publications, pulmonary NTM and new drug 136, pulmonary NTM and immunotherapy 13, pulmonary NTM and vaccine 46 publications. The database query resulted in a total of 197 publications. After the title and abstract review, 64 articles were included in this analysis.

### 2.1. Risk Factors of Pulmonary NTM

Patients with no identifiable structural lung diseases may develop pulmonary NTM but pulmonary NTM usually occurs in patients with underlying diseases [17,18,19,20]. Cystic fibrosis (CF) and chronic obstructive pulmonary disease (COPD) are common risk factors for pulmonary NTM. Other risk factors include old age, non-CF bronchiectasis, interstitial lung disease (ILD), and immunodeficiency [6,21,22].

CF is the most common genetic disorder in the US and Europe with incidence of about 1 in 3200 in Caucasians [23]. The incidence is much lower in other races. In the US, CF occurs in approximately 1:15,000 blacks, 1:35,000 individuals of Asian descent, and 1:10,900 Native Americans [24,25]. In US, there are more than 30,000 individuals with CF [26]. The number of adults with CF continues to increase, while the number of children remains relatively stable [26], because advances in medical care have prolonged life in these patients. With increases in survival the spectrum of life-threatening pulmonary infections in CF has changed. An extensive literature indicates that pulmonary NTM infections are becoming common in CF, with prevalence increasing from 10% in children aged 10 years to more than 30% in adults over 40 years [5]. Pulmonary NTM increases the rate of exacerbations in CF threefold [27], and accelerates deterioration of lung function [28,29,30] more than other serious CF pulmonary infections such as *Pseudomonas* and *Burkholderia* [29]. In addition, pre-transplant pulmonary NTM in CF patients makes post-lung transplant recovery more challenging [31]. It is likely that susceptibility to pulmonary NTM in CF is at least partly due to viscous secretions, deceased mucocilliary clearance, mucus tethering, and impaired innate immunity [32,33,34,35]. CF transmembrane conductance regulator (CFTR) gene modulator therapies have decreased rates of new or recurrent *Pseudomonas* lung infection and improved quality of life [36]. To our knowledge, there are no studies on the impact of CFTR modulator on pulmonary NTM. In fact, the prevalence of pulmonary NTM has continued to increase even after the wider use of CFTR modulators in the CF population [36]. CFTR modulators are drugs that act by improving production, intracellular processing, and/or function of the defective CFTR protein. The management may include two correctors and a potentiator (e.g., elexacaftor-tezacaftor-ivacaftor) to restore the function of the mutant CFTR protein more fully [37]. CFTR modulators may prevent lung damage or improve lung function but there is no evidence supporting reversal of already damaged bronchi, suggesting that CF patients with structural damage will continue to be high risk for pulmonary NTM. NTM infection significantly decreases lung function [30].

Chronic obstructive pulmonary disease (COPD) is another common risk factor for pulmonary NTM. The prevalence of (COPD) in the US adult population ranges from 5.1 to 14% [38,39,40,41]. Similar prevalence rates were reported from Canada and Europe with estimated global prevalence of 13% [42,43]. The prevalence of pNTM in COPD patients is about 0.7% with a hazard ratio of 15.5 for patients older than 35 years, with a 10× higher hazard ratio for patients ≥ 65 years old [4]. Pulmonary NTM doubles the rate of severe exacerbations requiring hospital admissions and leads to a significant deterioration in lung function [44]. In a population-based study of more than 6 million people, the fully adjusted hazard ratio for pulmonary NTM was significantly high in COPD (8.7, 95% CI 8.3–9.2) with incidence rate of 143 per 100,000 person-years compared to 6.6 per 100,000 person-years in a group without COPD [4].

### 2.2. Diagnosis of Pulmonary NTM

The diagnosis of pulmonary NTM relies on clinical, microbiologic, and radiologic criteria. Based on the guidelines by American Thoracic Society (ATS), European Respiratory Society (ERS), European Society of Clinical Microbiology and Infectious Diseases (ESCMID), and Infectious Diseases Society of America (IDSA), all of the following criteria have to be met for the diagnosis of pulmonary NTM [45,46]:New or worsening pulmonary symptoms with or without systemic symptoms.New or worsening radiologic findings suggestive of nodular, cavitary opacities on chest X-ray or bronchiectasis with nodules on computed tomography (CT).Exclusion of other diagnosis.Supportive microbiologic findings including (i) cultures of at least two separate sputum samples positive for NTM, (ii) culture of bronchial wash or lavage positive for NTM or (iii) lung histology showing granulomatous inflammation or acid-fast bacilli (AFB) and at least one positive NTM culture from biopsy or another respiratory specimen.

Productive cough and shortness of breath are the main presenting symptoms of pulmonary NTM, and some patients may have fatigue, fever, chest pain and weight loss [47,48]. Figure 1 shows typical radiologic findings of nodular and fibrocavitary pulmonary MAC

The ATS/ERS/ESCMID/IDSA criteria are based on expert opinion, not on high quality studies. Therefore, it may not be surprising to see some patients with underlying lung disease treated for pulmonary NTM even when they do not fulfill microbiologic (e.g., having only one culture positive sputum culture) or radiologic criteria (e.g., abnormal imaging study findings but not nodular, fibrocavitary or bronchiectasis) [48,49,50,51]. This may suggest the need to broaden the criteria for pulmonary disease caused by some of the NTMs. A retrospective study on 53 patients with and 19 without underlying lung diseases recommended the definition of definite pulmonary MAC for those who fulfill the ATS/ERS/ESCMID/IDSA criteria and probable pulmonary MAC for those with a single culture positive sputum or radiologic abnormalities other than nodular bronchiectasis or fibrocavitation [49]. The suggestion for classifying patients with pulmonary MAC into definite and probable groups must be tested on a large number of MAC patients with known treatment outcomes.

### 2.3. Clinical Relevance of Drug Susceptibility Testing

Macrolides, rifampin, ethambutol, and aminoglycosides are drugs used for treatment of pulmonary MAC. The first three drugs are used for all patients and an aminoglycoside is added for patients with fibrocavitary lesions [46]. In vitro susceptibility is not commonly done for new patients and only in vitro susceptibility for clarithromycin result has a good correlation with clinical efficacy [52]. Despite lack of correlation between in vitro susceptibility to some of the MAC drugs and clinical efficacy, it has been shown that high MIC particularly ≥8 mg/L for rifampin and ethambutol or ≥64 mg/L for aminoglycosides is associated with treatment failure [53,54]. In vitro testing using macrophages infected with MAC isolates from patients with disseminated MAC showed some promising results [55]. A decrease in colony forming units by at least one log was predictive of clinical efficacy of some of the drugs used for MAC [55].

The optimal drugs for treatment of pulmonary MAB are not known [46]. However, in one study that used combination regimen, treatment success was associated with in vitro susceptibility to clarithromycin but not with ciprofloxacin, doxycycline, cefoxitin or amikacin [56].

### 2.4. Treatment of Pulmonary MAC and MAB

In patients with pulmonary MAC, a susceptibility-based treatment for macrolides and amikacin is recommended by the ATS/ERS/ESCMID/IDSA guidelines, and it is suggested that patients with macrolide-susceptible MAC receive a treatment regimen with at least three drugs (including a macrolide and ethambutol) for at least 12 months after culture conversion [46]. For patients with cavitary, advanced/severe bronchiectasis or macrolide-resistant pulmonary MAC addition of amikacin or streptomycin is suggested [46]. The cure rates of treatment for pulmonary MAC ranges from 55% to 66% [48,57,58].

The optimal drugs, regimens, and duration of therapy for pulmonary MAB are not well defined. However, a susceptibility-based treatment for macrolides and amikacin is recommended [46]. Studies, mainly from South Korea, have shown that clarithromycin-containing oral regimen with 4–16 weeks of one or two intravenous drugs showed clinical improvement in 50–97% of patients [56,59,60]. In a retrospective study done on 65 patients in South Korea, efficacy of a combination regimen including clarithromycin, ciprofloxacin and doxycycline three-drug treatment with an initial 4-week course of intravenous cefoxitin and amikacin resulted in clinical improvement in 83% of patients, radiologic improvement in 74% and microbiologic cure in 58% of patients [56]. In this study, treatment for pulmonary MAB continued for at least 12 months after sputum culture conversion or a total duration of 24 months.

### 2.5. Prospects in Improving Diagnosis and Treatment

Microbiologic criteria for the diagnosis of pulmonary MAC and MAB include culture of respiratory specimen and histology [46]. In patients with only a single positive sputum culture or histology showing granuloma with neg cultures, new methods of detecting NTM infection may help in diagnosis. A dual skin test with MAC sensitin (MAS) and with Mtb purified protein derivative (PPD) was used for identification of patients with pulmonary MAC. MAS skin test with induration size of ≥5 mm larger than induration from PPD had a specificity of 97% for discriminating pulmonary MAC and TB [61].

Use of a simple and stable antigen for use in serodiagnosis has been explored. Glycopeptidolipids (GPL) have been the main focus because of their abundance in NTM isolates [62]. MAC strains produce highly antigenic and typeable serovar-specific GPLs. Observational study of anti-GPL on samples from 369 patients with rheumatoid arthritis, 10 with and 359 without pulmonary MAC showed a positive predictive value of 67% and a negative predictive value of 97%. Pulmonary MAC patients in this study fulfilled the criteria [63]. Using GPL from 11 reference strains of MAC in pulmonary MAC patients without rheumatoid arthritis, positive predictive values of 80% for IgG and 89% for IgA with corresponding negative predictive values of 89% and 97% [64].

Patients with cavitary, advanced/severe bronchiectasis or macrolide-resistant pulmonary MAC benefit from adding aminoglycosides as part of a treatment regimen [46]. In addition, aminoglycosdies are among the drugs to be considered in other pulmonary MAC patients who fail treatment. Unfortunately, use of intravenous or intramuscular aminoglycosides for extended duration is limited by serious side effects [43]. In the last few years, amikacin liposomal inhalation suspension (ALIS) has been shown to improve culture conversion in treatment refractory cases of pulmonary MAC [54,65,66]. The new guidelines now recommend the addition of ALIS in patients with pulmonary MAC pulmonary who have failed therapy after at least six months of guideline-based therapy [46].

Another approach to improve treatment outcome is finding ways to optimize the use of existing drugs. A multicenter study on efficacy of a single tablet with fixed dose combinations (FDC) of clarithromycin, rifabutin and clofazimine for treatment of pulmonary MAC with nodular bronchiectasis is underway [67]. FDC may prevent emergence of drug resistance but may not prevent treatment default associated with drug side effects. In a recent multicenter study of 297 pulmonary NTM patients, 90 (30.3%) required change in treatment because of drug side effects [48]. Therefore, simplifying treatment regimens to minimize the number of drugs helps improve treatment compliance. In a preliminary open-label study, a two-drug combination with clarithromycin and ethambutol was shown to be noninferior to a three-drug regimen with addition of rifampin [68]. This has the potential to decrease drug side effects and challenges arising from drug interactions. Currently a large randomized controlled trial is undergoing recruitment of 500 participants to study the benefit of two versus three-antibiotic therapy for MAC disease [69].

There are efforts to develop new drugs for MAC and MAB [70,71]. Part of this effort relies on testing new drugs developed for TB treatment [70,71]. There are encouraging results from studies that use cell cultures and small animal but not many have made it to human trials [72]. Potential candidates include gallium containing compounds. In mice, compounds containing gallium which interferes with iron metabolism or uptake by mycobacteria have been shown to have activity against pulmonary MAB [73]. A phase 1b study on intravenous gallium nitrate in CF colonized with NTM is underway [74].

A unique approach that showed some promising results is immunotherapy. In a small placebo-controlled intramuscular IFN-γ daily for one month and then three times per week for up to 6 months as adjunct to guideline-based therapy, a more rapid clinical and radiological responses were seen in patients who received IFN-γ [75]. Nitric oxide (NO), an important part of the innate immune system with bactericidal activities, has been tried for treatment of pulmonary MAB [76,77]. The use of inhaled NO by Yaacoby-Bianu et al. in 2018 in two cystic fibrosis patients with pulmonary MAB resulted in a reduction in bacterial load as measured by quantitative polymerase chain [78]. The use of inhaled NO by Goldbart et al. in a CF patient with pulmonary MAB showed bacterial growth inhibition as well as improvement of lung pathology on CT [79]. The observations from these case reports show that NO is safe and well tolerated and could play a crucial role in the treatment of NTM pulmonary disease [71,78,79]. Additional trials are investigating treatment outcomes with intermittent inhaled nitric oxide (NO) in cystic fibrosis versus non-cystic fibrosis patients, both with pulmonary MAB [80]. Another important immunomodulating therapy that attracted some interest is granulocyte-macrophage colony-stimulating factor (GM-CSF). GM-CSF is produced by different types of cells including T cells, macrophages and alveolar epithelial cells, and is known to increase the ability of macrophages to control mycobacterial infection [81]. Inhaled GM-CSF was tested in patients who failed MAC or MAB antibiotic treatment and helped achieve culture conversion in 5/32 (15.6%) but the culture conversion was durable after end of guideline-based therapy only in two patients [82]. There is still an interest to carefully examine the benefits of GM-CSF in treatment refractory pulmonary NTM cases, particularly MAB [83].

Immunotherapy with BCG has been considered. BCG, the only licensed mycobacterial vaccine, live attenuated *Mycobacterium bovis* is used globally for the prevention of pulmonary TB. BCG, is known to stimulate innate immunity [84,85,86] and induces adaptive immune responses [87]. Long term follow-up studies have demonstrated that BCG provides durable protection against pulmonary TB as summarized in two recent reviews [88,89]. BCG has also been shown to be effective in preventing infections due to other mycobacteria including *M. leprae*, *M ulcerans*, and strains of *M. avium* causing NTM lymphadenitis in children, indicating cross-protective immunity within the genus [90,91,92]. Our recent study showed that BCG induces cross-protective immunity to NTM [93]. The use of BCG as adjunct immunotherapy or prevention of pulmonary MAC and MAB is attractive. One potential advantage of whole cell vaccines over protein-adjuvant formulations and viral-vectored constructs is their broad antigen composition, which includes the complete protein repertoire, lipids, carbohydrates, and other moieties that may be antigenic and induce donor unrestricted T-cell responses, B-cell responses, and possibly also natural killer and innate lymphoid cell responses [94]. However, most patients with pulmonary MAC and MAB have underlying lung diseases and therefore, the use of BCG has a potential of causing serious BCG lung disease.

## 3. Conclusions

The prevalence of pulmonary NTM is increasing. The exact reasons for this increase in prevalence are not known, but medical advances to increase the lifespan of patients with underlying lung diseases could be one factor. Underlying lung diseases such as CF and COPD are major risk factors for pulmonary MAC and MAB. ATS/ERS/ESCMID/IDSA guidelines include recommendations for diagnosis and treatment of pulmonary MAC and MAB. Despite the guidelines, diagnosis of pulmonary MAC and MAB is challenging. Similarly, treatment of pulmonary MAC and MAB is complex with high failure rates. There is a need to improve the diagnosis and treatment of patients with pulmonary MAC and MAB. Efforts to simplify the treatment regimen for most cases of pulmonary MAC and findings new therapeutics are encouraging. Two-drug antibiotic regimens, inhaled amikacin, and immunotherapies have potential for the management of pulmonary NTM. 

## Figures and Tables

**Figure 1 microorganisms-11-00047-f001:**
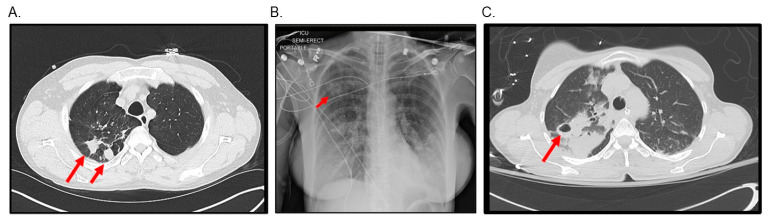
Radiologic abnormalities of patients with pulmonary MAC. (**A**) Chest CT of a 60-year-old female patient diagnosed of nodular pulmonary MAC. She completed treatment and remained culture negative for several months. (**B**,**C**), a chest X-ray and chest CT of a 60-year-old female patient recently diagnosed with fibrocavitary pulmonary MAC. Arrows show specific lesions.

## Data Availability

This review did not include data.

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
