# Peer review of "Diagnosis and Management of Pulmonary NTM with a Focus on Mycobacterium avium Complex and Mycobacterium abscessus: Challenges and Prospects"

_microorganisms, 2022, doi:10.3390/microorganisms11010047_

Round 1

Reviewer 1 Report

This review summarizes the latest useful findings from the literature on risk factors, diagnosis, drug susceptibility testing, and treatment advances in pulmonary MAC disease and pulmonary MAB disease. I think however that there are a few improvements that should be made before publication.

#1: Please indicate the period of publication of the article you are searching for.                     

#2: Pulmonary NTM disease is a risk factor for pre-existing lung diseases such as cystic fibrosis and COPD, but why not show that pulmonary NTM disease occurs even in the absence of pre-existing lung disease and is not infrequent?

#3: As discussed in this review, I think it is very important to raise the question of whether the diagnostic criteria of the current international guidelines are appropriate.

#4: For lines 139 to 134, the author should discuss whether there is a link between drug susceptibility(MIC) and clinical efficacy of drugs used to treat MAB.

# 5: For lines 162 to 166, I wonder if the utility of antibody testing for GPL is not limited to rheumatoid arthritis patients.

Author Response

We thank the reviewer for comments. The manuscript is revised based on the comments. See point-by-point response.

Reviewer 2 Report

This manuscript provides the reader a comprehensive review of the diagnosis and management of NTM, focusing specifically on 2 pathogens.

Please note my findings below:

1. Lines 48-49, rates of CF in USA and Europe are described solely for Caucasians. Reviewer suggests that other races be included to show a more broad inclusion of individuals. 

2. Line 62, please be more descriptive of CFTR therapies, including function and use.

3. Line 95, please define AFB.

4. In Figure 1, the radiologic modalities used need to be described in the legend, and for each of the 3 images. 

5. Line 162, please provide more details regarding antibodies to GPL and why its selection was important for diagnostic purposes.

6. Lines 210-211, just as with NO, please provide more detail regarding inhaled treatment with GM-CSF and why it failed as a therapy.

Author Response

We thank the reviewer for comments. All comments have been addressed in the revised manuscript.
